# Individual Immune Response to SARS-CoV-2 Infection—The Role of Seasonal Coronaviruses and Human Leukocyte Antigen

**DOI:** 10.3390/biology12101293

**Published:** 2023-09-28

**Authors:** Karla Rottmayer, Henry Loeffler-Wirth, Thomas Gruenewald, Ilias Doxiadis, Claudia Lehmann

**Affiliations:** 1Laboratory for Transplantation Immunology, University Hospital Leipzig, Johannisallee 32, 04103 Leipzig, Germany; 2Interdisciplinary Centre for Bioinformatics, IZBI, Leipzig University, Haertelstr. 16–18, 04107 Leipzig, Germany; 3Clinic for Infectious Diseases and Tropical Medicine, Klinikum Chemnitz, Flemmingstraße 2, 09116 Chemnitz, Germany

**Keywords:** SARS-CoV-2 immune response, coronavirus pandemic, individual SARS-CoV-2 antibody patterns, seasonal coronaviruses, vaccination, HLA association

## Abstract

**Simple Summary:**

The immune response towards SARS-CoV-2 during the recent pandemic was analyzed retrospectively in a cohort of 512 individuals comprising 334 females and 178 males. In total, 737 sera were tested against several proteins of the virus and six seasonal coronaviruses. Differential reactivity was observed, and different HLA proteins were found to be associated with the antibody response towards SARS-CoV-2. This allowed us to identify low- and high-responder individuals. These results are indicative for an individual host immune response and may enable future individualized protective measures.

**Abstract:**

During the coronavirus pandemic, evidence is growing that the severity, susceptibility and host immune response to SARS-CoV-2 infection can be highly variable. Several influencing factors have been discussed. Here, we investigated the humoral immune response against SARS-CoV-2 spike, S1, S2, the RBD, nucleocapsid moieties and S1 of seasonal coronaviruses: hCoV-229E, hCoV-HKU1, hCoV-NL63 and hCoV-OC43, as well as MERS-CoV and SARS-CoV, in a cohort of 512 individuals. A bead-based multiplex assay allowed simultaneous testing for all the above antigens and the identification of different antibody patterns. Then, we correlated these patterns with 11 HLA loci. Regarding the seasonal coronaviruses, we found a moderate negative correlation between antibody levels against hCoV-229E, hCoV-HKU1 and hCoV-NL63 and the SARS-CoV-2 antigens. This could be an indication of the original immunological imprinting. High and low antibody response patterns were distinguishable, demonstrating the individuality of the humoral response towards the virus. An immunogenetical factor associated with a high antibody response (formation of ≥4 different antibodies) was the presence of HLA A*26:01, C*02:02 and DPB1*04:01 alleles, whereas the HLA alleles DRB3*01:01, DPB1*03:01 and DB1*10:01 were enriched in low responders. A better understanding of this variable immune response could enable more individualized protective measures.

## 1. Introduction

The COVID-19 pandemic caused by the severe acute respiratory syndrome coronavirus type 2 (SARS-CoV-2) has influenced lives worldwide, caused 6.96 million deaths and left 771 million infected [1]. Therefore, understanding the pathomechanism and immune response towards the virus became a central endeavor of the scientific community. 

The immunity, being humoral (B cells, antibodies) and/or T-cell-mediated as part of the adaptive immunity, is currently in the scientific focus. The study of humoral antibody and cell-mediated immunity has been of particular interest given the pivotal role of adaptive immunity in protecting against viral infection and disease [2,3]. Several risk factors for susceptibility and severity of infection have been assessed, including age, gender, ethnicity and chronic diseases, such as diabetes or hypertension, as well as the virus variants and subvariants. [2]. In addition, the role of previous infections of severe respiratory disease with other human coronaviruses, such as MERS-CoV and SARS, was discussed. [4]. The endemic coronaviruses that cause mostly mild diseases are also known as human coronaviruses (hCoVs) 229E, HKU1, NL63 and OC43. Humans are exposed to these every year, and, as Edridge et al. [5] showed, the protective immunity is short-lasting and re-infections may occur annually. The homology of the spike proteins of the novel SARS-CoV-2 and the endemic hCoV exhibits about 30%, as shown by Hicks et al. [6]. This is raising the question of cross-immunity or pre-existing protective immunity, as reviewed by Galipeau et al. [4]. Various factors, such as age, gender, virus load and previous illnesses, influence the occurrence and outcome of the infection [2]. Furthermore, immunogenetic characteristics, which have an influence on the course and outcome of a SARS-CoV-2 infection, have been studied [7,8,9]. In particular, the human leukocyte antigen system (HLA) plays a crucial role in the human immune system. The HLA genes encode MHC (Major Histocompatibility Complex) molecules, which are cell surface proteins that link antigen binding and Lymphocyte activation. Two classes are distinguished: class I HLA genes, namely, HLA-A, HLA-B and HLA-C, encode proteins, which can present peptides and are recognized by CD8+ cytotoxic T cells. In contrast, HLA-II proteins are encoded by class II HLA genes HLA-DP, HLA-DQ and HLA-DR. HLA-II molecules are mainly found on antigen-presenting cells, which use them to present extracellular antigens to naive CD4+ T cells (Th 0). These T cells differentiate into T follicular helper cells (TFH), which in turn induce B cell differentiation in the germinal center of draining lymph nodes, eventually maturing into memory B cells and antibody-producing plasma cells [10,11]. However, several contradictory results were reported regarding the time course and anti-SARS-CoV-2 antibody levels. Gerhards et al. [12] and Wright et al. [13] report no substantial decrease in immunoglobulin G (IgG) levels over a period of 6 months. Vo et al. [14] showed a stable antibody level (in 88% of the individuals) up to 9 months after infection. The models of Loesche et al. [15] provide evidence of seroconversion up to 18 months after infection. A review by Kim et al. [16] reported stable nucleoprotein (N-)- and spike protein (S-)-specific IgG antibody titers for at least 4 months after diagnosis but also reported spike subunit 1 (S1) IgG levels that declined over 6 months. In relation to vaccination, Anderson et al. [17] found that SARS-CoV-2 S IgG antibody levels in primary vaccinated individuals declined steadily over an 11-month period and then rose above baseline levels with booster vaccination. Khoury et al. [18] also show that antibody titers peak after the second dose of vaccine and then decline rapidly over 150 days.

During our inclusion period, the first vaccination against COVID-19 was introduced in Germany on 27 December 2020 [19]. We used this date to divide the cohort into a natural infection phase until 27 December 2020 and a sample with longitudinal assessment of antibody response to immunization events. Most previous studies used enzyme-linked immunosorbent assays (ELISAs) to detect the antibodies [20,21,22]. Here, we focus on a multiplex flow cell technique, which has the advantage of detecting antibodies to different viral antigens simultaneously, providing insight into the pattern of reactivity to the different SARS-CoV-2 domains and other hCoVs, which could represent a part of the individual immune response [23,24,25]. Finally, we correlated the individual antibody responses with immunogenetic HLA patterns to analyze their influence on the immune response.

## 2. Materials and Methods

### 2.1. Cohort

In our study, 737 sera from 512 individuals collected in two Saxonian hospitals in Leipzig and Chemnitz were examined. They comprise 334 females and 178 males, aged 1 to 88 years. Blood samples were taken between April 2020 and February 2023. The study participants were asked whether they were infected with SARS-CoV-2 or not, and the date of SARS-CoV-2 PCR validating a previous infection was documented. Most of the infected persons had moderate disease severity. Only six subjects were hospitalized and received non-invasive ventilation. None of the infected persons required intensive care. From 2021, after vaccinations against COVID-19 became available, vaccination dates were additionally documented. The total cohort of this study was divided into two samples (Figure 1) (details are given in the next subchapters): (i) 379 persons that were measured during the natural infection phase, where vaccination was not available yet (collection April 2020–December 2020) and (ii) 278 persons with either SARS-CoV-2 infection or vaccination (collection April 2020–February 2023). Note that infected persons from the former sample are also contained in the latter one. The gender distribution of the overall cohort (about one-third males and two-thirds females; Figure 1a) is conserved in the two study samples, and the age distribution of the individuals also agrees well in them (Figure 1b).

#### 2.1.1. Natural Infection Phase—Sample

To study antibody responses to SARS-CoV-2, seasonal coronaviruses (sCoVs), MERS and SARS, the natural infection phase sample collects sera of persons until 27 December 2020, when SARS-CoV-2 vaccinations became available. This sample was divided into two groups: The first group comprises 183 people with PCR-confirmed SARS-CoV-2 infection (108 female and 75 male; mean age 52.2 years). The second group collects 196 people without prior SARS-CoV-2 infection (145 female and 51 male; mean age 44.2 years). 

#### 2.1.2. Immunization—Sample

In the second sample, we investigate antibody responses to SARS-CoV-2 after the first immunization event. Here, 278 individuals were enrolled, of which 266 (161 females, 105 males) had a single infection as their first immunization event. A second group with 12 persons (7 females, 5 males) received a single vaccination as the first immunization event. The mean age was 48.7 years in both groups. 

### 2.2. Longitudinal Antibody Data

A total of 31 individuals provided multiple sera during the collection period (15 females, 16 males; mean age 47.0 years. Time points of this longitudinal data were categorized according to a sequence of defined events: 1—after SARS-CoV-2 infection, 2—before the 1st vaccination, 3—after the 1st vaccination, 4—before the 2nd vaccination, 5—after the 2nd vaccination, 6—before the 3rd vaccination, 7—after the 3rd vaccination, 8— before the 4th vaccination, and 9—after the 4th vaccination.

To evaluate differences between the vaccinated individuals with or without a prior infection, we accordingly defined a ‘vaccination after infection’ group (11 individuals, 5 female and 6 male) and a ‘vaccination without prior infection’ group (20 individuals, 10 female and 10 male). Figure 2 shows the time of blood collection starting with day 0 being the day of infection or first sampling date, respectively, and the corresponding time point categories.

### 2.3. Multiplex Bead-Based Immunoassay

Antibodies were tested using the LabScreen™ COVID Plus assay (One Lambda, West Hills, CA, USA), and the mean fluorescence intensity (MFI) was then measured using the Luminex™ 200™ instrument system according to the manufacturer’s recommendation (Luminex™, Corp., Austin, TX, USA), as described in [23]. The test kit contains 13 different antigenetic determinants attached to polystyrene beads: five antigens specific for SARS-CoV-2, namely, the full spike protein, three subunits of the spike protein: the S1 domain, the receptor-binding domain (RBD) and the S2 domain, and the nucleocapsid protein (NCP). In addition, six endemic coronavirus antigens were simultaneously tested against the S1 domains of the four endemic human coronaviruses hCoV-NL63, hCoV-HKU1, hCoV-229E and hCoV-OC43 and the severe human coronaviruses SARS-CoV and MERS-CoV. A positive and negative control were included to complement the panel. The positive control bead is coated with human IgG and the negative control bead is coated with human albumin. With this assay, we were able to test up to 95 different sera in a single run. MFI values were determined using xPONENT v4.3 software and evaluated and normalized with Fusion v 4.5 software (Luminex, Corp., Austin, TX, USA). 

For qualitative analyses, the manufacturer’s cut-off values for each bead were used to classify positive and negative antibody formation.

For the quantitative SARS-CoV-2 antibody analyses, we calculated the International Units using a regression with a dilution series of the WHO International Standard, which will be discussed in a validation paper (manuscript in preparation).

### 2.4. Dichotomization into High and Low Responders with Regard to SARS-CoV-2 Infection

To investigate the role of HLA genes in the immune response to SARS-CoV-2 infection, the 266 infected patients from the immunization sample were assessed for the number of antibodies formed against the SARS-CoV-2 full spike protein, spike subunit 1 domain (S1), receptor-binding domain (RBD), spike subunit 2 domain (S2) and nucleocapsid protein (NCP). Antibodies were considered ‘formed’ when corresponding mean fluorescence intensity (MFI) exceeded the cut-off values assigned by the manufacturer (One Lambda, West Hills CA, USA). Individuals who formed four or five different SARS-CoV-2 antibodies were defined as high responders; individuals with up to three antibodies were defined as low responders. In total, 202 high responders (58% female) and 68 low responders (68% female) were defined this way, with respect to the 5 tested SARS-CoV-2-domains.

### 2.5. HLA Typing with Next-Generation Sequencing (NGS)

After the DNA isolation from the EDTA blood samples according to manufacturer’s recommendation (QIAamp DNA Blood Mini Kit, QIAGEN, Hilden, Germany), the HLA typing was performed as described in [8] using two commercial test kits: the Alltype NGS 11-Loci (One Lambda, West Hills, CA, USA) and AlloSeq^®^ Tx 17 (CareDx, San Francisco, CA, USA).

### 2.6. Statistical Analyses

The Wilcoxon rank-sum test was used to compare antibody levels between different groups, such as infected persons versus non-infected and vaccination after infection versus without infection. 

Enrichment analyses of HLA alleles in the high and low responders was performed using Fisher’s exact test. 

All statistical analyses were performed using R 4.2.2 (RStudio 2022.12.0 Build 353).

### 2.7. Ethical Approval

Our study complied with the Helsinki Declaration and was approved by the Ethics Committee of the Medical Faculty of the University of Leipzig. We received a signed declaration of consent from each patient (195/20-ek, 20 May 2020).

## 3. Results

### 3.1. Humoral Response against sCoVs, MERS-CoV, SARS-CoV and SARS-CoV-2 in the Natural Infection Phase without Vaccinations

#### 3.1.1. Responses against sCoVs, MERS-CoV and SARS-CoV

Our first aim is to trace immune responses to seasonal coronaviruses, MERS-CoV and SARS-CoV. For this, we compare antibody levels against hCoV-229E, hCoV-HKU1, hCoV-NL63, hCoV-OC43, MERS-CoV and SARS-CoV in persons that experienced a SARS-CoV-2 infection to those in persons without infection (Figure 3 and Appendix A). Corresponding blood samples of 379 persons in total were taken in the natural infection phase of our study. 

In both groups, over 90% of the patients showed antibodies against hCoV-229E (95%; 91%) and against hCoV-NL63 (95%; 90%). The antibody levels of hCoV-HKU1 and hCoV-OC43 were positive in over 85% of both infected (89%; 88%) and non-infected patients (89%; 85%). However, very few patients showed a positive antibody response to MERS-CoV regardless of SARS-CoV-2 infection status (6%; 2%). Only the antibody response to SARS-CoV differed between the two groups markedly. In the PCR positive groups, 74% formed antibodies, whereas the non-infected patients had a positive antibody response in only 15%. 

As expected, MERS-CoV antibodies were present in only a few individuals as this virus is restricted mainly to the Arabian Peninsula. The antibody response against seasonal coronaviruses was high, as expected, with a small increase caused by infection with SARS-CoV-2. SARS-CoV and SARS-CoV-2 are closely related, which explains the marked increase in antibodies against SARS-CoV after SARS-CoV-2 infection.

After qualitative evaluation of antibody formation, we now compare quantitative MFI levels of the antibody beads between infected and non-infected individuals (Figure 4). Interestingly, we observed higher MFI levels in the non-infected group for hCoV-229E, hCoV-HKU1, and hCoV-NL63 compared to the individuals with a previous SARS-CoV-2 infection (*p*-values < 0.001 in Wilcoxon rank-sum test). The level of the hCoV-OC43 antibody did not differ significantly in the two groups. Probably, the antigen–antibody reaction plateau was reached, so that a further MFI increase was not possible due to the limited hCoV-OC43 antigen provided. This is also supported by general lower MFI levels against hCoV-OC43 compared to the others viral antigens. The measured MFI levels towards the MERS-CoV antigens were low in both groups, as expected, but significantly higher in the infected individuals (*p*-value < 0.001). Interestingly, antibody levels for SARS-CoV were higher in SARS-CoV-2-infected persons (*p*-value < 0.001), in contrast to the seasonal coronaviruses. This finding underlines a potential cross-immunity between SARS-CoV and SARS-CoV-2.

#### 3.1.2. Response against SARS-CoV-2

Now we examined antibody levels against different SARS-CoV-2 domains in the natural infection phase to judge latent immunization in the cohort and to compare between the non-infected and the infected group. We found that the vast majority of the individuals in the SARS-CoV-2-infected group had detectable antibodies against all SARS-CoV-2 domains (Figure 5); however, the lowest response in this group was observed against the nucleocapsid protein with 70%. In the non-infected group, only a few patients (7–9%) had detectable antibodies against the SARS-CoV-2 beads tested. We can conclude that a latent immunization against SARS-CoV-2 can be assumed for about 5–10% of the cohort population. 

#### 3.1.3. Antibody Formation against the SARS-CoV-2 Domains Reveals Immuno-Reactive Patterns

Next, we want to conduct a complementary, pattern-wise view of antibody formation. For this, we generate patterns of five ‘+’ or ‘−‘ signs, depending on the formation of antibodies against the five SARS-CoV-2 epitope domains. Different antibody patterns emerged in the natural infection phase sample (Figure 6). Most naïve individuals (*n* = 171) did not show an antibody response to any SARS-CoV-2 antigen. However, there are also 22 formerly SARS-CoV-2-infected individuals who did not develop antibodies. In contrast, most of the infected individuals (*n* = 114) developed antibody responses to all of the SARS-CoV-2 antigens. The third most frequent pattern is a response to all antigens except the nucleocapsid protein. This pattern was found in 17 infected and one non-infected individual. This is due to the waning of NCP antibodies post-infection. Ten infected individuals reacted only to the full spike, the RBD and the S2 domain. Five individuals without a positive PCR test developed a positive antibody reaction to the S1 domain. A positive antibody response to all antigens except the S1 protein was seen in four infected and one non-infected individuals. Another pattern is the reaction to all antigens except the full spike antigen, which was found in four individuals with a PCR-proven infection. Three infected and one non-infected individuals reacted only to the full spike, S2 and NCP domain. Other patterns were found in 17 individuals (Figure 6). 

#### 3.1.4. Correlation Analysis between Antibody Levels against Seasonal Coronaviruses, MERS-CoV, SARS-CoV and SARS-CoV-2

The correlations between the antibody MFI profiles of the 379 individuals in the natural infection phase sample against the sCoVs, MERS-CoV, SARS-CoV and the new SARS-CoV-2 can be seen as an indicator for cross-immunization (Figure 7). The closest relations among the endemic coronaviruses are between hCoV-NL63 and hCoV-229E as well as hCoV-OC43 and MERS (see dendrogram and correlation coefficients in Figure 7a and Figure 7b, respectively). Between the SARS-CoV-2 antigens, we observed a high correlation between the full spike and the S2 domain as well as between the receptor-binding domain and the S1 domain. The correlation matrix splits into two triangular matrices, one with high correlations within the SARS-CoV-2 antigens and one within the seasonal coronaviruses. As expected, there is a strong correlation between SARS-CoV-2 and SARS-CoV. Consistent with the low epitope homology, MERS-CoV correlates only with itself. Interestingly, there are moderate negative correlations between SARS-CoV-2 antigens and sCoV antigens. For example, high levels of MFI in hCoV-NL63 are associated with lower MFI levels of SARS-CoV-2 and vice versa. 

### 3.2. Humoral Response towards SARS-CoV-2 after the First Immunization Event

In our immunization sample, we analyzed sera from individuals after their first immunization event, namely, either a SARS-CoV-2 infection or vaccination. Due to the large difference in case numbers between the groups (266 had a single infection as their first immunization event vs. only 12 persons received a single vaccination as the first immunization event), any subsequent analysis could only give an indication. Similar to the results of infected individuals in the natural-phase sample, most individuals formed antibodies to all SARS-CoV-2 antigens. In response to the first vaccination, most individuals formed antibodies to all SARS-CoV-2 antigens except NCP. Detailed results are available in the Appendix A. 

### 3.3. Longitudinal Analysis of Antibody Levels against SARS-CoV-2

Since the end of 2020, individuals have provided multiple serum samples over the course of the study. A detailed list of collection dates is shown in Figure 2. We divided this longitudinal sample into two groups: people that received vaccination after a prior SARS-CoV-2 infection and people who were vaccinated without a known previous infection. In both groups, antibody levels against the spike domains increased with vaccination and then decreased over time and subsequently rose again with each follow-up vaccination (see Figure 8a for S1 subunit and Appendix A for the other domains). Antibodies exceeded the cut-off of 1.45 IU/mL throughout the study period. Antibody levels against the nucleocapsid protein behaved differently. They were only detectable after infection and then fell below the cut-off of 1.71 IU/mL within about 100 days until time point 2 (1st vaccination; Figure 8b). In the group without prior SARS-CoV-2 infection, antibodies against the nucleocapsid protein were below the cut-off for the entire period. The individual course of each domains’ antibody levels is shown in Appendix A.

Comparison of the two groups before the first vaccination was given (time point 2) showed that significantly higher antibody levels were measured in the group with a preceding infection: These differences are highly significant for the full spike, S1, S2 and receptor-binding domains (*p*-values < 0.01 in Wilcoxon rank-sum test; Figure 8a and Appendix A). The IU/mL values for the nucleocapsid protein were lower in the group with no pre-vaccination infection (*p*-values < 0.01; Figure 8b).

After the first vaccination, the IU/mL measured in those with a previous infection were higher against the full spike (*p*-value < 0.05), S1 and S2 domains (*p*-value < 0.01 both) and RBD (*p*-value < 0.01). There was no significant difference in antibody levels against the NCP after the first vaccination (Figure 8a and Appendix A), because the vaccines used did not include NCP epitopes.

### 3.4. HLA Enrichment Analysis of Low and High Responders

Finally, we aim to extract HLA alleles, which are associated with strong or weak immune responses to SARS-CoV-2 infection, respectively. Therefore, we grouped persons who experienced a SARS-CoV-2 infection according to the number of antibodies formed against the SARS-CoV-2 full spike protein, S1 domain, RBD, S2 domain and NCP: High responders are individuals with antibodies against four or all of the five virus domains; low responders form three or less antibodies only. The gender and age distributions are shown in Figure 9.

We found enrichment of the HLA alleles A*26:01, C*02:02 and DPB1*04:01 in the high responder group. In contrast, the HLA alleles DRB3*01:01, DPB1*03:01 and DPB1*10:01 were enriched in the low responders, as presented in Figure 10.

## 4. Discussion

In the first part, we investigated the humoral immune response to all humanpathogenic coronaviruses. Our data show a high seroprevalence to seasonal coronaviruses (hCoV-229E, hCoV-HKU1, hCoV-NL63 and hCoV-OC43). In 80–90% of individuals in both the infected and non-infected groups, antibodies were detectable. This confirms results published in the review by Huang et al. [26], who looked at age-related seroprevalence. In adults, the highest seroprevalence was found for hCoV-229E (around 80%), followed by hCoV-OC43 and hCoV-NL63 (around 70%), and the lowest for hCoV-HKU1 (25 to 75%). Low seroprevalence is expected for MERS-CoV and SARS-CoV due to their epidemiology. MERS-CoV arose from a zoonotic spillover in the Arabian Peninsula in 2012. A review by Grant et al. [27] described a seroprevalence of 0.1% in general population samples. It is not clear if the few individuals in our cohort who formed antibodies to the MERS-CoV antigen (6% in the infected group and 2% in the non-infected group) were exposed to the virus while travelling or had a broad cross-reactive B cell response to coronavirus antigens. For SARS-CoV, which occurred in 2002, Lueng et al. [28] reported a seroprevalence of 0.1% in a meta-analysis. However, our data show that 15% of the subjects in the non-infected group tested positive for antibodies to the SARS-CoV antigen. The difference between the two studies is most likely caused by the cross-reactivity of SARS-CoV and SARS-CoV-2 antibodies in those subjects, who were SARS-CoV-2-infected without a positive PCR and were therefore misclassified as non-infected. This could also be the reason for the positive antibody formation against SARS-CoV-2 antigens found in the non-infected group. In particular, asymptomatic individuals may not have been SARS-CoV-2 PCR tested. The high cross-reactivity between SARS-CoV and SARS-CoV-2 antibodies may also be the reason for the 70% of people in the infected group with a positive antibody response against SARS-CoV. When comparing MFI levels for each seasonal coronavirus between those with and without a previous SARS-CoV-2 infection, higher levels were seen in the non-infected group for hCoV-229E, hCoV-HKU1 and hCoV-NL63. This could be due to a possible protective effect of higher antibody levels on susceptibility to SARS-CoV-2 infection, which is able to inhibit SARS-CoV-2 entry into the host cell as described by Ng et al. [29]. Other studies have also shown that previous infections with seasonal coronaviruses are associated with less severe SARS-CoV-2 infections [30,31]. In contrast, Lin et al. [32] found no difference in antibody levels to sCoVs in SARS-CoV-2-infected and non-infected individuals and therefore concluded that there is no evidence for protection by previous sCoV infections. Using the multiplex antibody detection method, correlations between different antigens were examine in more detail. The dendrogram shows that the closeness of the relationships is consistent with the familial relationships described in [33] (Figure 7a). The *alphacoronaviridae* hCoV-229E and hCoV-NL63 show a strong positive correlation (Spearman r = 0.83). A further strong positive correlation was found for the *betacoronaviridae* hCoV-OC43 and hCoV-HKU1 (Spearman r = 0.65). Interestingly, we found moderate negative correlations between antibodies to SARS-CoV-2 antigens and sCoV antigens. This may be due to a decrease in antibodies within the first 20 days after infection, as reported by Lin et al. [32]. Another explanation could be the effect of the original immunological imprinting. Memory B cells are prone to produce more antibodies against the sCoVs and suppress the SARS-CoV-2-specific antibody formation [4,34]. In contrast, several studies have discussed a back-boost of antibodies against sCoVs after SARS-CoV-2 infection as a sign of memory B cell reactivity [33,35,36,37]. In addition, there is a strong positive correlation between all SARS-CoV-2 antigens and the SARS-CoV S1 antigen, particularly with the S2 domain of SARS-CoV-2 (Spearman r = 0.78). Cross-reactivities are reported for the test we used [23] as well as in other studies [3,33,38]. Furthermore, both viruses share more than 70% homology [2] and the S2 subunit is the most conserved domain. Therefore, this may be the reason for the strong correlation. [29] Analysis of immune-reactive patterns revealed the individuality of the humoral immune response to SARS-CoV-2 infection. Most infected individuals are able to form all antibodies, but there are also individuals with PCR-proven infection in whom no antibodies were detected. A possible explanation for this is that the antibodies are not yet detectable or are no longer detectable, depending on the kinetics of the antibodies. Seroconversion can be observed within the first and third week after PCR confirmation [12,39,40], and stable antibody levels have been reported up to nine months after infection [12,13,14]. In our cohort, the median time between PCR confirmation and sampling is 70 days (IQR 23.75, 149). Interestingly, once any antibody is detectable, antibodies to the S2 domain are almost always present. Antibodies to the full spike antigen and RBD are frequently detected. Not much has been published about those antibody patterns, with the exception that NCP antibodies are more rarely detectable compared to antibodies to all other SARS-CoV-2 antigens [41,42]. Similar patterns of reactivity occur after the first immunization event (Appendix A). When considering the antibodies to the first immunization event, individuals showed good antibody formation to each SARS-CoV-2 antigen after natural infection. Regarding the vaccinated group, individuals showed good antibody formation to each SARS-CoV-2 antigen except against NCP. The vaccines that were used comprised mRNA vaccines [43,44], which induce a transient expression of the SARS-CoV-2 spike antigen, and a replication-deficient chimpanzee adenovirus (ChAdOx1) vector vaccine encoding the SARS-CoV-2 S glycoprotein [45]. After this initial immune response, we were able to follow the formation of antibodies in some individuals over time. Since the vaccination campaign fell within our observation period, the resulting data had to be considered in this regard. Limited by the non-uniform sampling intervals, the following statements can be made: Antibody levels to the spike antigens rise with vaccination and then fall over the time, only to rise again with subsequent vaccinations, but they remain above the detection limit throughout the period. This behavior was also reported previously [17]. Khoury et al. showed a peak in RBD antibody levels after the second vaccination and a rapid decline over a period of 4 months [18]. Antibodies to NCP are detectable only after infection and then fall rapidly below the detection limit. In comparable terms, Kannenberg et al. showed an antibody half-life of NCP IgG of 119 days over a period of one year compared to the RBD IgG half-life of 183 days. The decline was particularly rapid in the first 200 days with a T_1/2_ of NCP IgG of 88 days [46]. In solely vaccinated individuals, antibodies against NCP were below the cut-off for the entire period, which is again consistent with the mechanism of action of the vaccines. To find an explanation of why the immune response is so individual, we looked at the highly polymorphic HLA alleles present in our cohort. To observe whether enrichments result in a broader or narrower immune response, the cohort was divided into those individuals with up to 3 antibodies formed against SARS-CoV-2 antigens (low responder) and those who showed a broad immune response with ≥4 antibodies formed (high responder). We found enrichments in the high responder group for the HLA alleles A*26:01, C*02:02 and DPB1*04:01 and enrichment in the low responder group for the HLA allele DPB1*03:01. We hypothesized that the HLA class II alleles enriched in the high responders may have a greater affinity to bind the SARS-CoV-2 peptides in the HLA molecules on the antigen-presenting cells and thus induce a more vigorous activation of cytotoxic T cells and B cells, leading to antibody production. Many different studies on HLA association have been published so far, differing from our results [10,47,48]. This may be due to the study population and differences in the sample size and methodology.

## 5. Conclusions

The present study shows that individuality influences the immune response towards the SARS-CoV-2 virus. Among this, the HLA set up of the individuals seems to influence the immune response positively and negatively. However, other polymorphic immune-related components, such as cytokines, probably have additive effects. In addition, it might be of interest how each variant of concern (VOC) may induce a different immune response. In future studies, including these additional factors might help to understand the immune response and provide information for individually helping affected patients. 

## Figures and Tables

**Figure 1 biology-12-01293-f001:**
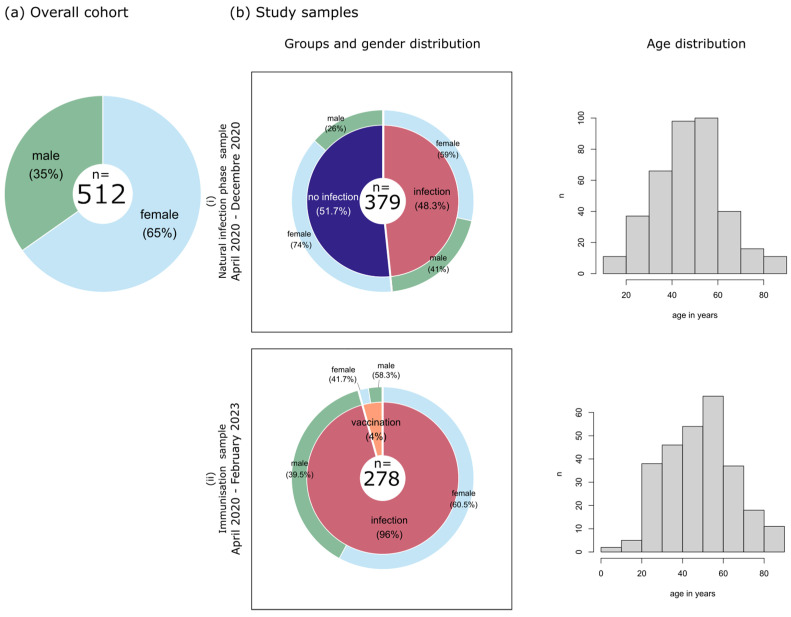
Characterization of the study cohort: (**a**) Distribution of genders in the total cohort reveals that majority are female. (**b**) Our analyses involve two study samples derived from this overall cohort: (**i**) persons measured in the natural infection phase and (**ii**) persons with SARS-CoV-2 immunization due to either infection or vaccination. Note that time frames as well as persons overlap between these two samples. Gender and age distributions are shown for both samples.

**Figure 2 biology-12-01293-f002:**
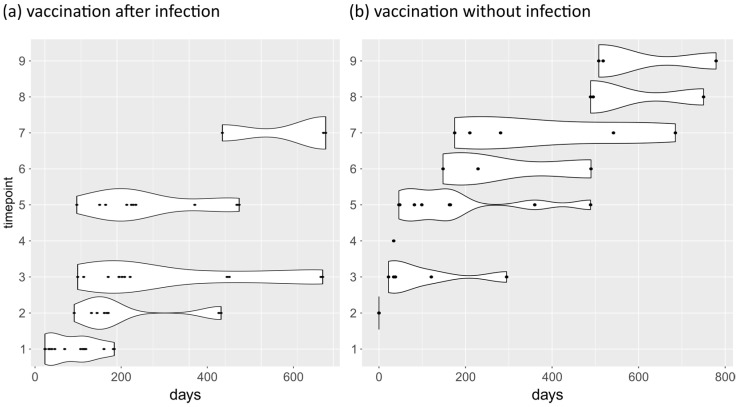
Blood collection dates in relation to immunization events. (**a**) For the ‘vaccination after infection’ group, the days since the first positive SARS-CoV-2 PCR test results are shown. (**b**) For the ‘vaccination without prior infection’ group, the starting time was set to the first blood sampling date before first vaccination. Labeling of the Y-axis: 1—after infection, 2—before the 1st vaccination, 3—after the 1st vaccination, 4—before the 2nd vaccination, 5—after the 2nd vaccination, 6—before the 3rd vaccination, 7—after the 3rd vaccination, 8—before the 4th vaccination, 9—after the 4th vaccination.

**Figure 3 biology-12-01293-f003:**
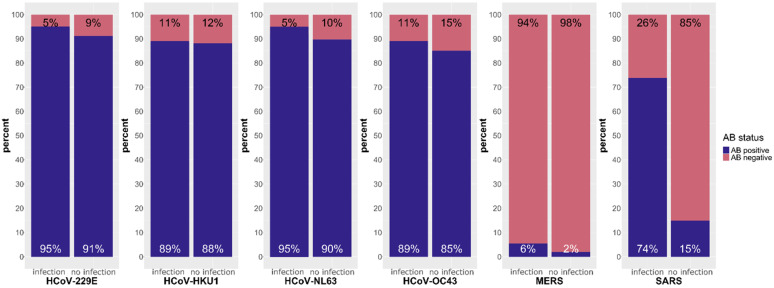
Fractions of persons with and without prior SARS-CoV-2 infection forming antibodies towards the sCoVs hCoV-229E, hCoV-HKU1, hCoV- NL63 and hCoV-OC43, MERS and SARS. Left columns represent the individuals with a previous SARS-CoV-2 infection and the right columns represent the individuals with no detected SARS-CoV-2 infection, respectively. Within each column, the blue bars indicate proportion of individuals with a positive antibody reaction and the rose bars indicate antibody reactions below the cut-offs.

**Figure 4 biology-12-01293-f004:**
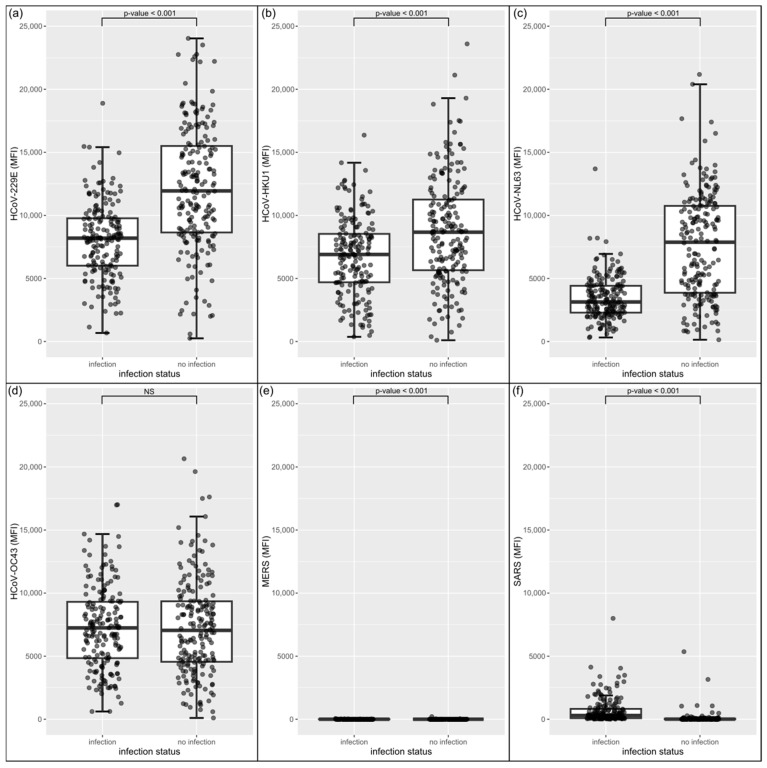
Comparison of MFI levels towards seasonal coronaviruses, MERS and SARS: (**a**) hCoV-229E, (**b**) hCoV-HKU1, (**c**) hCoV-NL63, (**d**) hCoV-OC43, (**e**) MERS, (**f**) SARS. The left boxplot represents the individuals with a previous SARS-CoV-2 infection and the right boxplot the individuals with no detected SARS-CoV-2 infection.

**Figure 5 biology-12-01293-f005:**
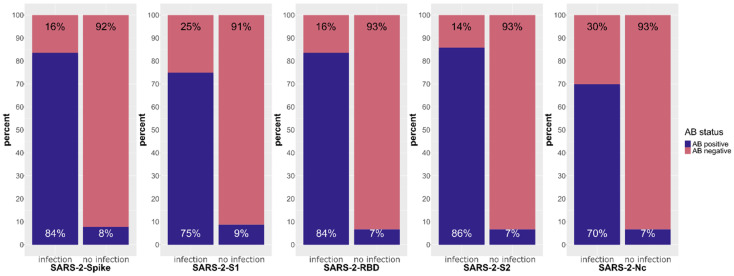
Fractions of persons with positive and negative antibody reactions towards the SARS-CoV-2, full spike, S1 domain, receptor-binding domain (RBD), S2 domain and SARS-CoV-2 nucleocapsid protein. See description of Figure 3.

**Figure 6 biology-12-01293-f006:**
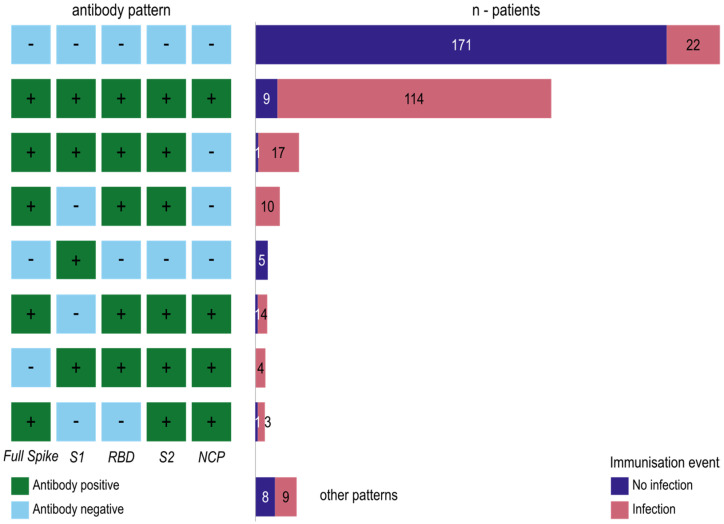
Antibody patterns against SARS-CoV-2 domains full spike, S1 domain, receptor-binding domain, S2 domain and nucleocapsid protein. On the left side, the SARS-CoV-2 immuno-pattern is shown, where ‘+’ signs and green boxes indicate formation of antibodies against the corresponding domain and ‘−‘ signs and blue boxes indicates missing antibodies. The right side shows the number of individuals that feature a particular immuno-pattern, divided into those with a previous SARS-CoV-2 infection and those without. Only patterns present in at least four individuals are shown in this figure.

**Figure 7 biology-12-01293-f007:**
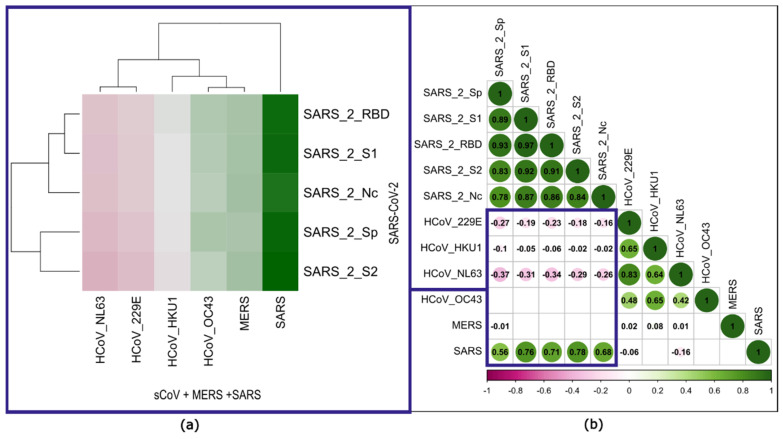
Spearman correlation analysis between antibody levels against the endemic coronaviruses and the new SARS-CoV-2. (**a**) The clustered heatmap presents the correlation between antibodies against the endemic coronaviruses (as columns) and the SARS-CoV-2 domains (as rows). The more intense the pink color, the stronger the negative correlation. Vice versa, the more intense the green color, the stronger the positive correlation. (**b**) The dotplot shows all pairwise correlation values computed from the endemic and the SARS-2 coronavirus antigen levels. The numbers in the middle of the circles are showing the Spearman correlation coefficients, and the sizes of the circles scale with absolute correlation.

**Figure 8 biology-12-01293-f008:**
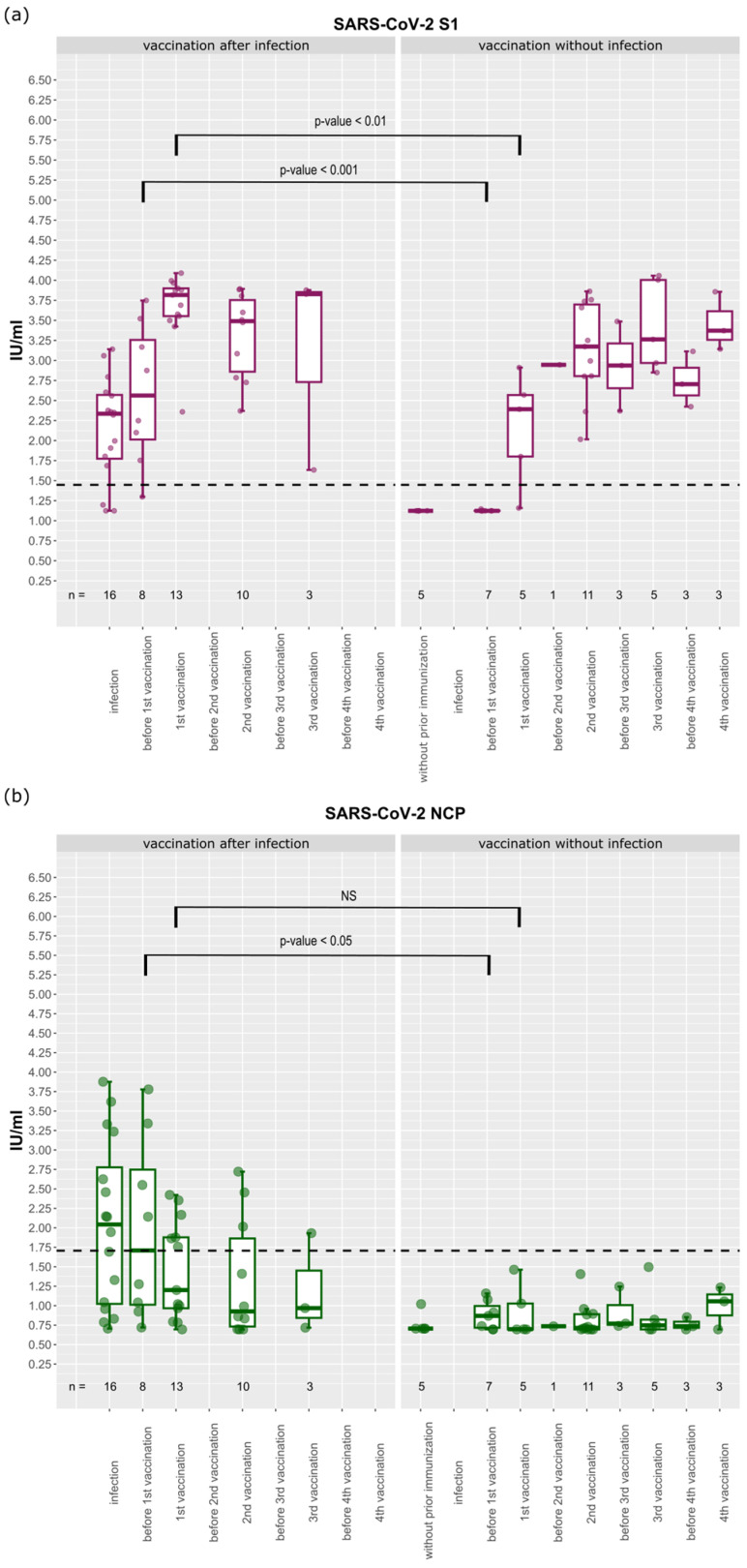
Longitudinal development of the antibody levels (**a**) against the SARS-CoV-2 S1 domain and (**b**) against the nucleocapsid protein. The other domains are shown in Appendix A. The boxplots on the **left side** indicate the antibody levels of individuals with previous SARS-CoV-2 infection and subsequent vaccination. The boxplots on the **right side** indicate individuals who were only vaccinated. The dashed lines mark the detection cut-offs, which differ for each domain.

**Figure 9 biology-12-01293-f009:**
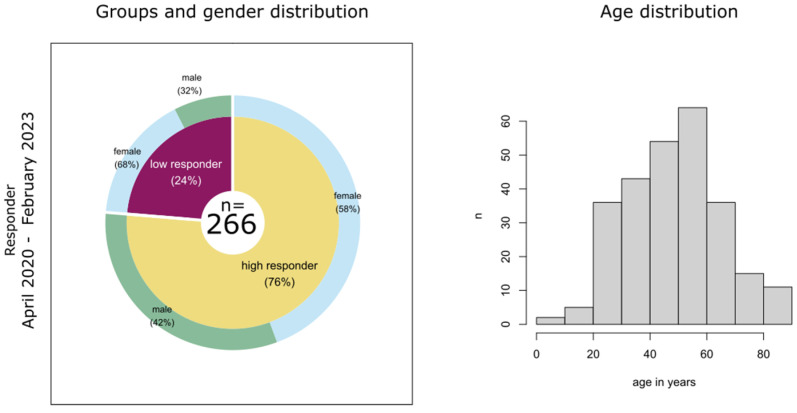
Gender and age distributions in the high and low responders (divided by their reaction to the SARS-CoV-2 infection).

**Figure 10 biology-12-01293-f010:**
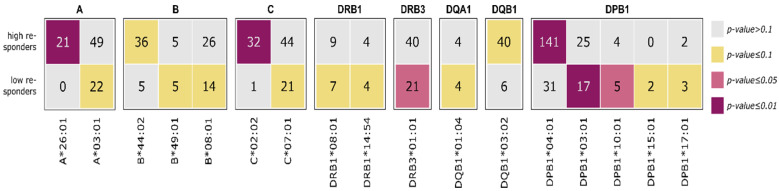
Contingency tables of HLA class I and II alleles enriched in high and low SARS-CoV-2 antibody responders, respectively. Numbers refer to all individuals with the respective allele, divided into high and low responders. Colors indicate the significance of the enrichment according to Fisher’s exact test (see figure legend). Only alleles with *p*-value < 0.1 are shown.

## Data Availability

The data supporting this study’s findings are available from the corresponding authors upon reasonable request.

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
