# Peer review of "Individual Immune Response to SARS-CoV-2 Infection—The Role of Seasonal Coronaviruses and Human Leukocyte Antigen"

_biology, 2023, doi:10.3390/biology12101293_

Round 1

Reviewer 1 Report

1) Between 2020 to 2023, during which samples were collected, individuals may have encountered multiple variants of the original SARS-2 virus.  Has the author account for variability due to infection with different SARS-2 strains?

2) It is unclear at what time samples from the "2.1.2 Immunization - sample" cohort were taken after 1st immunization.  When the samples were taken, did the individual had only a single vaccination/single confirmed infection?

3) Also in the "2.1.2 Immunization - sample" cohort, Figure 9 and Figure 10, there is too much sample number discrepancy for any confident conclusion to be drawn in comparisons between the 266 infected-as-1st-immunization vs only 12 vaccination-as-1st-immunization taken after 1st immunization.  Authors should clearly indicate this in the text.

4) Same with the longitudinal antibody data - too few samples to confidently draw any strong conclusion from.  Authors should clearly indicate this in the text.

5) Reference to Figure 4 is missing

6) Line 243 - should this be Figure 5 and not Figure 4?

7) Figure 4 - MERS (MFI) graph - distribution is difficult to observe due to the scaling of the y-axis, which makes is hard to confirm the authors statement that MFI-levels are "significantly higher in the infected individuals...".  Can the authors modify to permit better presentation of this data?

8) Figure 6 and Figure 10 - mis-spelling of graph label for infection group

9) Lines 327, 330, and 391 - grammatical correction needed

10) Figure 10 - are the infected samples analysed here the same as Figure 6?

11) Line 470 - "soVs"...should it be 'sCoVs'?

12) Line 108: "...natural infection phase, were vaccination was not available", should be "where" vaccination was not available.

Author Response

We would like to express our sincere thanks for the corrections and constructive suggestions for the submitted manuscript.

1) Our results are limited by the lack of information on the variant of concern of the observed infections. We mentioned this point in the conclusion.

2) Corrected line 127-130 to: „Here, 278 individuals were enrolled, of which 266 (161 females, 105 males) had a single infection as their first immunization event. A second group with 12 persons (7 females, 5 males) received a single vaccination as the first immunization event. The mean age was 48.7 years in both groups.“

3) Under the subsection 3.2: Due to the large difference in case numbers between the groups (266 had a single infection as their first immunization event vs only 12 persons received a single vaccination as the first immunization event), any subsequent analysis could only give an indication. 

4) Sample size n were added in Figure 8

5) Added in subsection 3.1.1.: "After qualitative evaluation of antibody formation, we now compare quantitative MFI levels of the antibody beads between infected and non-infected individuals (Figure 4)."

6) Corrected to Figure 5

7) To ensure comparability of the individual bead-specific antibody levels, we have retained the scaling of the figure.  The text refers to the low MERS antibody levels.: ”The measured MFI-levels towards the MERS-CoV antigens were low in both groups as expected, but significantly higher in the infected individuals (p-value<0.001).”

8) Corrected

9) Revised:

Line 327 now in the supplement: In particular, we see very similar antibody levels in both groups for all antigens, except for the nucleocapsid protein, where antibodies are only formed in those individuals who have been infected.

Line 330 now in the supplement:  The qualitative analysis shown in Figure 8 is similar to quantitative results obtained by transformation of the results into IU/ml (Figure S3).

Line 391 now line 320-322: A detailed list of collection dates is shown in Figure 2. We divided this longitudinal sample into two groups: people that received vaccination after a prior SARS-CoV-2 infection and people who were vaccinated without a known previous infection.

10) No, there is a difference in the time of sample collection: Figure 6 shows data from the natural phase cohort (sample dates until December 2020) with 183 infected persons. On the other hand in Figure 10 data from the immunization cohort (sample dates until February 2023) are shown with 266 infected persons.  

11) Line 470 now line 402: Corrected to sCoVs

12) Line 108: Corrected to „natural infection phase, where vaccination was not available yet“

Kind regrads

The authors

Reviewer 2 Report

Rottmayer and colleagues have tested the antibody level against human CoVs in SARS-CoV-2 infected or vaccinated individuals during the COVID-19 pandemic. This manuscript show that the SARS-2 antibody level is negatively related to seasonal CoVs and positively related to SARS-CoV. Antibody response patterns were highly correlated to some HLA alleles. This study provides information about the SARS-CoV-2 infection/vaccination caused immune response as well as its correlations to other human CoVs. However, there are some questions should be further addressed.    

1. It seems that seasonal COVs antibody level was significantly influenced after SARS-2 infection, authors may add more discussion about the potential mechanism.

2. How to explain the SARS-2 antibody positive results in SARS-2 uninfected patients? And also the MERS-CoV antibody positive result? Is it the real infection events or false positive result?

3. Authors should further describe the standard for dividing antibody positive/negative.

Authors mentioned that the antibody pattern was correlated with some HLA loci. However, other structural proteins such as E, M and non-structural proteins were not concluded in this study. thus, it’s not appropriate to simply divide high or low response patients by using > 4 types of antibodies as an indicator. Authors may further analyze the correlation between antibody level and HLA loci.

4. due to the limited samples size in vaccination individuals, and the poor correlation with the main content, the size of 3.2 section can be decreased.

5.There are too many separated figures in the main text, some similar figures can assort into one figure.

6. Check the figure number cited in line 291.

no

Author Response

We would like to express our sincere thanks for the corrections and constructive suggestions for the submitted manuscript. Here are the point-by-point answers:

1.) With our study design, it is not possible to analyse a possible effect of SARS-CoV-2 infection on sCoV antibody levels, as we do not have preliminary data on individuals. However, we refer to other studies in our discussion: “In contrast, several studies have discussed a back-boost of antibodies against sCoVs after SARS-CoV-2 infection as a sign of memory B-cell reactivity [33, 35-37].”

2.) Added to the discussion (line 390-392): “This could also be the reason for the positive antibody formation against SARS-CoV-2 antigens found in the non-infected group. In particular, asymptomatic individuals may not been SARS-CoV-2 PCR tested.”

3.) Line 161-162:For qualitative analyses, the manufacturer's cut-off values for each bead were used to classify positive and negative antibody formation.

The dichotomization into high and low responders refers to the ability of an immune response, the ability to produce different antibodies in the 5 SARS-CoV-2 domains tested, not to the antibody levels.

Due to the imprecision of the antibody levels with our semi-quantitative measurement method, we only correlate the qualitative ability to form different antibodies to the HLA-alleles. This may give an indication of the antigen presentation capacity of the HLA encoded MHC molecules.

Added line 177: “….in respect to the 5 tested SARS-CoV-2-domains.”

4.) section 3.2. moved to the supplement

5.) Fewer figures in the main text because of the removal of section 3.2.

6.) Corrected to 7a and b

Kind regards

The authors